# THOMPSON SAMPLING FOR (COMBINATORIAL) PURE EXPLORATION

## ABSTRACT

Pure exploration plays an important role in online learning. Existing work mainly focuses on the UCB approach that uses confidence bounds of all the arms to decide which one is optimal. However, the UCB approach faces some challenges when looking for the best arm set under some specific combinatorial structures. It uses the sum of upper confidence bounds within arm set $S$ to judge whether $S$ is optimal. This sum can be much larger than the exact upper confidence bound of $S$, since the empirical means of different arms in $S$ are independent. Because of this, the UCB approach requires much higher complexity than necessary. To deal with this challenge, we explore the idea of Thompson Sampling (TS) that uses independent random samples instead of the upper confidence bounds to make decisions, and design the first TS-based algorithm framework TS-Verify for (combinatorial) pure exploration. In TS-Verify, the sum of independent random samples within arm set $S$ will not exceed the exact upper confidence bound of $S$ with high probability. Hence it solves the above challange, and behaves better than existing UCB-based algorithms under the general combinatorial pure exploration setting. As for pure exploration of classic multi-armed bandit, we show that TS-Verify achieves an asymptotically optimal complexity upper bound.

## 1 INTRODUCTION

Pure exploration is an important task in online learning, and it tries to find out the target arm as fast as possible. In pure exploration of classic multi-armed bandit (MAB) (Audibert et al., 2010), there are totally $m$ arms, and each arm $i$ is associated with a probability distribution $D_i$ with mean $\mu_i$. Once arm $i$ is pulled, it returns an observation $r_i$, which is drawn independently from distribution $D_i$ by the environment. At each time step $t$, the learning policy $\pi$ either chooses an arm $i(t)$ to pull, or chooses to output an arm $a(t)$. The goal of the learning policy is to pull arms properly, such that with an error probability at most $\delta$, its output arm $a(t)$ is the optimal arm (i.e., $a(t) = \arg\max_{i \in [m]} \mu_i$) with complexity (the total number of observations) as low as possible.

Pure exploration is widely adopted in real applications. For example, in the selling procedure of cosmetics, there is always a testing phase before the commercialization phase (Audibert et al., 2010). The goal of the testing phase is to help to maximize the cumulative reward collected in the commercialization phase. Therefore, instead of regret minimization (Berry & Fristedt, 1985; Auer et al., 2002), the testing phase only needs to do exploration (e.g., to investigate which product is the most popular one), and wants to find out the target with both correctness guarantee and low cost. In real world, sometimes the system focuses on the best action under a specific combinatorial structure, instead of the best single arm (Chen et al., 2014). For example, a network routing system needs to search for the path with minimum delay between the source and the destination. Since there can be an exponential number of paths, the cost of exploring them separately is unacceptable. Therefore, people choose to find out the best path by exploring the single edges, and this is a pure exploration problem instance in combinatorial multi-armed bandits (CMAB). In this setting, we still pull single arms (base arms) at each time step, and there is a super arm set $\mathcal{I} \subseteq 2^{[m]}$. The expected reward of a super arm $S \in \mathcal{I}$ is $\sum_{i \in S} \mu_i$, i.e., the sum of the expected rewards of its contained base arms. And the goal of the player is to find out the optimal super arm with an error probability at most $\delta$.

Most of the existing solutions for pure exploration follow the UCB approach (Audibert et al., 2010; Kalyanakrishnan et al., 2012; Chen et al., 2014; Kaufmann & Kalyanakrishnan, 2013). They com-

pute the confidence bounds for all the arms, and claim that one arm is optimal only if its lower confidence bound is larger than the upper confidence bounds of all the other arms. Though this approach is asymptotically optimal in pure exploration of classic MAB problems (Kalyanakrishnan et al., 2012; Kaufmann & Kalyanakrishnan, 2013), it faces some challenges in the CMAB case (Chen et al., 2014). In the UCB approach, for a super arm $S$, the algorithm usually uses the sum of upper confidence bounds of all its contained base arms as its upper confidence bound. This means that the gap between the empirical mean of super arm $S$ and the used upper confidence bound of $S$ is about $\tilde{O}(\sum_{i \in S} \sqrt{1/N_i})$ (here $N_i$ is the number of observations on base arm $i$). However, since the observations of different base arms are independent, the standard deviation of the empirical mean of super arm $S$ is $\tilde{O}(\sqrt{\sum_{i \in S} 1/N_i})$, which is much smaller. This means that the used upper confidence bound is much larger than the real one. Combes et al. (2015) deal with this problem by computing the upper confidence bounds for all the super arms independently, which leads to an exponential time cost and can be hard to implement. In fact, existing efficient UCB-based solutions either suffer from a higher complexity bound (Chen et al., 2014; Gabillon et al., 2016), or need further assumptions on the combinatorial structure to achieve an optimal complexity bound (Chen et al., 2017).

Then a natural solution to deal with this challenge is to use random sample of arm $i$ (with mean to be its empirical mean and standard deviation to be $\tilde{O}(\sqrt{1/N_i})$) instead of its upper confidence bound to judge whether a super arm is optimal. If we let the random samples of different base arms be *independent*, then the gap between the empirical mean of super arm $S$ and the sum of random samples within $S$ is $\tilde{O}(\sqrt{\sum_{i \in S} 1/N_i})$, which has the same order as the gap between the empirical mean of super arm $S$ and its real mean. Therefore, using independent random samples can behave better than using confidence bounds, and this is the key idea of Thompson Sampling (TS) (Thompson, 1933; Kaufmann et al., 2012; Agrawal & Goyal, 2013). In fact, many prior works show that TS-based algorithms have smaller cumulative regret than UCB-based algorithms in regret minimization of CMAB model (Wang & Chen, 2018; Perrault et al., 2020). However, existing researches only study using TS approaches under the Bayesian setting of pure exploration (Li et al., 2021), and there still lack results of adapting TS to the frequentist setting.

In this paper, we attempt to fill up this gap, and study using TS in pure exploration under the frequentist setting for both MAB and CMAB instances by adapting the idea of TS in the Bayesian setting (Li et al., 2021). We emphasize that it is non-trivial to design (and analyze) a TS-based algorithm under the frequentist setting. The first challenge is that there is a lot more uncertainty in the random samples than the confidence bounds. In UCB-based algorithms, the upper confidence bound of the optimal arm is larger than its expected reward with high probability. Thus, the arm with the largest upper confidence bound is either an insufficiently learned sub-optimal arm (i.e., the number of its observations is not enough to make sure that it is a sub-optimal arm) or the optimal arm, which means that the number of pulls on sufficiently learned sub-optimal arms is limited. However, for TS-based algorithms, there is always a constant probability that the random sample of the optimal arm is smaller than its expected reward. In this case, the arm with the largest random sample may be a sufficiently learned sub-optimal arm. Therefore, if we still choose to pull the arm with the largest random sample (similar as UCB-based policies that pull the arm with the largest upper confidence bound), then the mechanism of the TS-based policy should be designed carefully to make sure that we can still obtain an upper bound for the number of pulls on sufficiently learned sub-optimal arms.

Another challenge is that TS is an algorithm that follows the Bayesian framework and loses many good properties in the frequentist setting. In the Bayesian setting, at each time step $t$, the parameters of the game follow a posterior distribution $\mathcal{P}_t$, and the random samples are drawn from $\mathcal{P}_t$ independently as well. Therefore, using the random samples to output the optimal arm can explicitly ensure the correctness of the TS-based algorithm. However, in the frequentist setting, the parameters of the game are fixed but unknown, and they have no such correlations with the random samples. Because of this, if we still use the random samples to output the optimal arm, then the distributions of random samples in the TS-based algorithm need to be chosen carefully to make sure that it still has correctness guarantee in the frequentist setting.

Besides, the analysis of TS-based algorithms in pure exploration is also very different with regret minimization. In regret minimization, at each time step, we only need to draw *one* sample for each base arm. However, in pure exploration, one set of samples is not enough at each time step, since the algorithm needs to i) check whether there is an arm that is the optimal one with high probability; ii) look for an arm that needs exploration, and none of these two goals can be achieved by only *one*

set of samples. Therefore, we must draw several sets of samples to make decisions, and this may fail existing analysis in regret minimization. For example, the analysis in (Agrawal & Goyal, 2013) obtains a lower bound for the probability that the optimal arm has the highest sample in *one* sample set. However, when we use several sets of samples to make decisions, we can notice the optimal arm only if it has the highest sample in *most* of these sample sets, which is much harder than it has the highest sample in only *one* sample set when its observations are bad (e.g., its empirical mean equals to its real expected reward minus the confidence radius).

In this paper, we solve the above challenges, and design a TS-based verification framework TS-Verify for (combinatorial) pure exploration under the frequentist setting. TS-Verify takes a target (super) arm as input, and aims to verify that this target (super) arm is optimal with error constraint $\delta$. At each time step $t$, TS-Verify first draws independent random samples $\theta_i^k(t)$ for all the (base) arms $i \in [m]$ and $k = 1, 2, \cdots, M$ (i.e., totally $M$ independent samples for each arm). Then it tries to find out the $M$ best (super) arms under sample sets $\boldsymbol{\theta}^k(t) = [\theta_1^k(t), \cdots, \theta_m^k(t)]$ for $k = 1, 2, \cdots, M$. If in most of these sample sets, the target (super) arm is the best one, then the algorithm will output that the target (super) arm is indeed optimal. Otherwise, the algorithm will choose to pull a (base) arm from the exploration set, which contains the target (super) arm and all the (super) arms that have been optimal in at least one sample set at this time step.

In the rest of this paper, we mainly focus on the behaviour (i.e., correctness guarantee and complexity upper bound) of this verification algorithm, which is almost the same as a complete pure exploration algorithm after we adapt the explore-then-verify framework (Karnin, 2016; Chen et al., 2017). In the general CMAB setting, we show that TS-Verify is near optimal and behaves better than existing UCB-based algorithms (Chen et al., 2014). The optimal algorithm (Chen et al., 2017) requires the combinatorial structure to satisfy some specific properties, which is less general than our results. As for the MAB case, we show that TS-Verify is asymptotically optimal, i.e., it has a comparable complexity as existing optimal algorithms (Kalyanakrishnan et al., 2012; Kaufmann & Kalyanakrishnan, 2013) when $\delta \to 0$. All these results indicate that out algorithm framework is efficient and general in dealing with pure exploration problems. To the best of our knowledge, this is the first result of using TS-based algorithms in pure exploration under the frequentist setting.

## 2  RELATED WORKS

Pure exploration of the classic MAB model is first proposed by Audibert et al. (2010). After that, people have designed lots of learning policies for this problem. The two most representative algorithms are successive-elimination (Even-Dar et al., 2006; Audibert et al., 2010; Kaufmann & Kalyanakrishnan, 2013) and LUCB (Kalyanakrishnan et al., 2012; Kaufmann & Kalyanakrishnan, 2013). Both of them adapt the idea of UCB (Auer et al., 2002) and achieve an asymptotically optimal complexity upper bound (i.e., it matches with the complexity lower bound proposed by Kalyanakrishnan et al. (2012)). Compared to these results, our TS-Verify policy uses a totally different approach, and can achieve an asymptotically optimal complexity upper bound as well.

Combinatorial pure exploration is first studied by Chen et al. (2014). They propose CLUCB, an LUCB-based algorithm that is efficient as long as there is an offline oracle to output the best super arm under any given parameter set. Chen et al. (2017) then design an asymptotically optimal algorithm for this problem. However, they require the combinatorial structure to follow some specific constraints so that they can apply a more powerful offline oracle. Recently, based on the game approach, Jourdan et al. (2021) provide another optimal learning policy for pure exploration in CMAB. But their algorithm still requires the existence of a more powerful offline oracle. Compared with these UCB-based algorithms, our TS-based algorithm achieves a better complexity bound than CLUCB (Chen et al., 2014) in the most general setting. However, since we do not assume the existence of a more powerful offline oracle, we cannot achieve the optimal complexity upper bound.

## 3  PRELIMINARIES

### 3.1  PURE EXPLORATION IN MULTI-ARMED BANDIT

A pure exploration problem instance of MAB is a tuple $([m], \boldsymbol{D}, \delta)$. Here $[m] = \{1, 2, \cdots, m\}$ is the set of arms, $\boldsymbol{D} = \{D_1, D_2, \cdots, D_m\}$ are the corresponding reward distributions of the arms,

and $\delta$ is the error constraint. In this paper, we assume that all the distributions $D_i$'s are supported on $[0,1]$. Let $\mu_i \triangleq \mathbb{E}_{X \sim D_i}[X]$ denote the expected reward of arm $i$, and $a^* = \arg\max_{i \in [m]} \mu_i$ is the optimal arm with the largest expected reward. Similar with many existing works (e.g., (Audibert et al., 2010)), we also assume that the optimal arm is unique. At each time step $t$, the learning policy $\pi$ can either pull an arm $i(t) \in [m]$, or output an arm $a(t) \in [m]$. If it chooses to pull arm $i(t)$, then it will receive an observation $r_{i(t)}(t)$, which is drawn independently from distribution $D_{i(t)}$. The goal of the learning policy is to make sure that with probability at least $1 - \delta$, its output $a(t) = a^*$. Under this constraint, it aims to minimize the complexity $Z^\pi \triangleq \sum_{i=1}^m N_i(T^\pi)$, where $T^\pi$ denotes the time step $t$ that policy $\pi$ chooses to output $a(t)$, and $N_i(t)$ denotes the number of observations on arm $i$ until time step $t$.

Let $\Delta_{i,m} \triangleq \mu_{a^*} - \mu_i$ denote the expected reward gap between the optimal arm $a^*$ and any other arm $i \neq a^*$. For the optimal arm $a^*$, its $\Delta_{a^*,m}$ is defined as $\mu_{a^*} - \max_{i \neq a^*} \mu_i$. We also define $H_m \triangleq \sum_{i \in [m]} \frac{1}{\Delta_{i,m}^2}$, and existing works (Kalyanakrishnan et al., 2012) show that the complexity lower bound of any pure exploration algorithm is $\Omega(H_m \log \frac{1}{\delta})$.

### 3.2 PURE EXPLORATION IN COMBINATORIAL MULTI-ARMED BANDIT

A pure exploration problem instance of CMAB is an extension of the MAB case. The arms $i \in [m]$ are called base arms, and there is also a super arm set $\mathcal{I} \subseteq 2^{[m]}$. For each super arm $S \in \mathcal{I}$, its expected reward is $\sum_{i \in S} \mu_i$. Let $S^* = \arg\max_{S \in \mathcal{I}} \sum_{i \in S} \mu_i$ denote the optimal super arm with the largest expected reward, and we assume that the optimal super arm is unique (as in (Chen et al., 2014)). At each time step $t$, the learning policy $\pi$ can either pull a base arm $i(t) \in [m]$, or output a super arm $S(t) \in \mathcal{I}$. The goal of the learning policy is to make sure that with probability at least $1 - \delta$, its output $S(t) = S^*$. Under this constraint, it also wants to minimize its complexity $Z^\pi$.

As assumed in many existing works (Chen et al., 2013; 2014; Wang & Chen, 2018), we also assume that there exists an offline Oracle, which takes a set of parameters $\boldsymbol{\theta} = [\theta_1, \cdots, \theta_m]$ as input, and outputs the best super arm under this parameter set, i.e., $\mathsf{Oracle}(\boldsymbol{\theta}) = \arg\max_{S \in \mathcal{I}} \sum_{i \in S} \theta_i$.

In this paper, for $i \notin S^*$, we use $\Delta_{i,c} \triangleq \sum_{j \in S^*} \mu_j - \max_{S \in \mathcal{I}: i \in S} \sum_{j \in S} \mu_j$ to denote the expected reward gap between the optimal super arm $S^*$ and the best super arm that contains $i$. As for $i \in S^*$, its $\Delta_{i,c}$ is defined as $\Delta_{i,c} \triangleq \sum_{j \in S^*} \mu_j - \max_{S \in \mathcal{I}: i \notin S} \sum_{j \in S} \mu_j$, i.e., the expected reward gap between $S^*$ and the best super arm that does not contain $i$. We also define width $\triangleq \max_{S \neq S^*}(|S \setminus S^*| + |S^* \setminus S|)$, and let $H_{1,c} \triangleq \text{width} \sum_{i \in [m]} \frac{1}{\Delta_{i,c}^2}$, $H_{2,c} \triangleq \text{width}^2 \sum_{i \in [m]} \frac{1}{\Delta_{i,c}^2}$.

Chen et al. (2017) prove that the complexity lower bound is $\Omega(H_{0,c} \log \frac{1}{\delta})$, where $H_{0,c}$ is the solution of the following optimization problem (here $\Delta_{S,c} = \sum_{i \in S^*} \mu_i - \sum_{i \in S} \mu_i$):

$$\min \quad \sum_{i \in [m]} N_m$$

$$\text{s.t.} \quad \sum_{i \in S \setminus S^*} \frac{1}{N_i} + \sum_{i \in S^* \setminus S} \frac{1}{N_i} \leq \Delta_{S,c}^2, \qquad \forall S \in \mathcal{I}, S \neq S^*$$

The following result shows the relationships between $H_{0,c}$, $H_{1,c}$ and $H_{2,c}$.

**Proposition 1.** *For any combinatorial pure exploration instance, $H_{0,c} \leq H_{1,c} \leq H_{2,c}$.*

*Proof.* Since width $\geq 1$, we have that $H_{1,c} \leq H_{2,c}$. As for the first inequality, note that $\forall i \in [m]$, $N_i = \frac{\text{width}}{\Delta_{i,c}^2}$ is a feasible solution of the optimization problem. Hence we have that $H_{0,c} \leq H_{1,c}$. $\square$

### 3.3 GENERAL EXPLORE-THEN-VERIFY FRAMEWORK

Due to space limit, the detailed descriptions of the general explore-then-verify framework (Chen et al., 2017) is deferred to Appendix A.

**Fact 1.** *(Lemma 4.8 in (Chen et al., 2017)) If there exist i) a complete pure exploration algorithm that outputs the optimal arm with probability at least $1 - \delta$ and complexity at most $O(H_E \log \frac{1}{\delta} + C_E)$;*

---

**Algorithm 1** TS-Verify

---

1: **Input:** Error constraint $\delta$, target $\tilde{a}$ (or $\tilde{S}$), $q \in [\delta, 0.1)$, $t = m$, $N_i = 0$, $R_i = 0$ for all $i \in [m]$.
2: Pull each arm once, update their number of pulls $N_i$'s and the sum of their observations $R_i$'s.
3: **while true do**
4:   **for** $k = 1, 2, \cdots, M(\delta, q, t)$ **do**
5:     For each arm $i$, draw sample $\theta_i^k(t)$ independently from distribution $\mathcal{N}(\frac{R_i}{N_i}, \frac{C(\delta, q, t)}{N_i})$.
6:     $a^k(t)$ ($S^k(t)$) is the best (super) arm under parameter set $[\theta_1^k(t), \cdots, \theta_m^k(t)]$.
7:   **end for**
8:   $\hat{p}_{\tilde{a}}(t) = \frac{1}{M(\delta, q, t)} \sum_{k=1}^{M(\delta, q, t)} \mathbb{I}[a^k(t) = \tilde{a}]$ ($\hat{p}_{\tilde{S}}(t) = \frac{1}{M(\delta, q, t)} \sum_{k=1}^{M(\delta, q, t)} \mathbb{I}[S^k(t) = \tilde{S}]$).
9:   **if** $\hat{p}_{\tilde{a}}(t) > 1 - q$ ($\hat{p}_{\tilde{S}}(t) > 1 - q$) **then**
10:     **Return:** Arm $\tilde{a}$ (super arm $\tilde{S}$) is indeed the optimal one.
11:   **else**
12:     Arbitrarily choose $a' \in \{a^k(t)\}_{k=1}^{M(\delta, q, t)}$ such that $a' \neq \tilde{a}$ (arbitrarily choose $S' \in \{S^k(t)\}_{k=1}^{M(\delta, q, t)}$ such that $S' \neq \tilde{S}$).
13:     Pull arm $i(t)$ from the exploration set $\{a', \tilde{a}\}$ ($\{S', \tilde{S}\}$), update its number of pulls $N_{i(t)}$ and the sum of its observations $R_{i(t)}$.
14:     $t \leftarrow t + 1$.
15:   **end if**
16: **end while**

---

*and ii) a verification algorithm such that with probability at least $1 - \delta$, it outputs that its input is optimal with complexity at most $O(H_V \log \frac{1}{\delta} + C_V)$ if its input is the optimal arm and does not terminate otherwise. Then we can construct a complete pure exploration algorithm based on the general explore-then-verify framework. It works correctly with probability at least $1 - \delta$, and its expected complexity is upper bounded by $O(H_E \log \frac{1}{\delta_0} + H_V \log \frac{1}{\delta} + C_E + C_V)$ for fixed $\delta_0 = 0.1$.*

According to Fact 1, in the rest of this paper, we focus on designing verification algorithms based on the TS approach. As long as we have a verification algorithm with error probability $\delta$ and complexity upper bound $O(H_V \log \frac{1}{\delta} + C_V)$, using any efficient complete pure exploration algorithm (e.g., LUCB (Kalyanakrishnan et al., 2012) or CLUCB (Chen et al., 2014)) in the explore-then-verify framework can make sure that the expected complexity of the constructed pure exploration algorithm is upper bounded by $O(H_E \log \frac{1}{\delta_0} + H_V \log \frac{1}{\delta} + C_E + C_V)$, which is asymptotically the same as the complexity upper bound $O(H_V \log \frac{1}{\delta})$ (since all the other terms do not depend on $\delta$).

## 4 THOMPSON SAMPLING-BASED VERIFICATION

### 4.1 GENERAL VERIFICATION FRAMEWORK FOR PURE EXPLORATION

Our Thompson Sampling-based verification framework (TS-Verify) is described in Algorithm 1. We use $N_i$ to denote the number of observations on arm $i$, $R_i$ to denote the sum of all the observations from arm $i$, and $N_i(t), R_i(t)$ to denote the value of $N_i, R_i$ at the beginning of time step $t$.

At each time step $t$, for any $i \in [m]$, TS-Verify first draws $M(\delta, q, t) \triangleq \frac{\log(12t^2/\delta)}{q^2}$ random samples $\{\theta_i^k(t)\}_{k=1}^{M(\delta, q, t)}$ independently from a Gaussian distribution with mean $\hat{\mu}_i(t) \triangleq \frac{R_i(t)}{N_i(t)}$ (i.e., the empirical mean of arm $i$) and variance $\frac{C(\delta, q, t)}{N_i(t)}$ (i.e., inversely proportional to $N_i(t)$), where $C(\delta, q, t)$ is a term depending on the settings and will be explained in detail later. Then TS-Verify checks which (super) arm is optimal in the $k$-th sample set $[\theta_1^k(t), \cdots, \theta_m^k(t)]$ for all $k$. In the MAB case, it can directly check which arm's sample is the largest one; in the CMAB case, it can use Oracle to obtain the best super arm under the $k$-th sample set. After that, TS-Verify checks whether the number of sample sets in which the target arm $\tilde{a}$ (or target super arm $\tilde{S}$) is the optimal one is large enough. If the number of such sample sets is larger than $(1 - q)M(\delta, q, t)$, then TS-Verify outputs that the target (super) arm is optimal. Otherwise it knows that either the target (super) arm is not optimal, or some (super) arms are not sufficiently learned (i.e., they do not have enough observations). In this case, TS-Verify constructs an exploration set $\{a', \tilde{a}\}$ (or $\{S', \tilde{S}\}$), where $a' \neq \tilde{a}$ is an arbitrary arm

(or $S' \neq \tilde{S}$ is an arbitrary super arm) that has been optimal in at least one sample set, and uses some choosing rules (which will be explained in detail later) to pull an arm $i(t)$ in the exploration set.

The reason that we use Gaussian random samples in TS-Verify is because it can simplify the analysis significantly (since the sum of independent Gaussian random variables is still a Gaussian one). We believe that using other kinds of random samples (e.g., uniform distributions in the confidence intervals) will have a similar behaviour.

## 4.2 TS-Verify for Multi-armed Bandit

We first apply TS-Verify to the classic MAB case. In the following, we define $\Phi(x, \mu, \sigma^2) \triangleq \Pr_{X \sim \mathcal{N}(\mu, \sigma^2)}[X \geq x]$, and $\phi(x)$ satisfies that $\Phi(\phi(x), 0, 1) = x$ for all $x \in (0, 1)$.

In the MAB case, we set $C(\delta, q, t) = \frac{4 \log(12mt^2/\delta)}{\phi^2(3q)}$, and for the exploration set $\{a', \tilde{a}\}$, we will choose to pull the arm with a smaller number of observations.

**Theorem 1.** *For $q \in [\delta, 0.1)$, with probability at least $1 - \delta$, we have that: i) if $\tilde{a} = a^*$, TS-Verify will return that arm $\tilde{a}$ is the optimal arm with complexity upper bounded by $O(H_m(\log \frac{1}{\delta} + \log(mH_m))\frac{\log \frac{1}{\delta}}{\log \frac{1}{q}} + H_m \frac{\log^2(mH_m)}{\log \frac{1}{q}})$; ii) if $\tilde{a} \neq a^*$, TS-Verify will not stop. Specifically, if we choose $q = \delta$, then the complexity upper bound is $O(H_m \log \frac{1}{\delta} + H_m \log^2(mH_m))$.*

**Remark 1.** *The value $q$ in TS-Verify is used to control the number of times that we draw random samples at each time step. Note that $M(\delta, q, t) = \frac{\log(12t^2/\delta)}{q^2}$. Hence when $q$ becomes larger, we need fewer samples, but the complexity bound becomes worse. Here is a trade-off between the algorithm's complexity and the number of random samples it needs to draw. Our analysis shows that using $q = \delta^{\frac{1}{\beta}}$ for some constant $\beta \geq 1$ can make sure that the complexity upper bound remains the same order, and reduce the number of random samples significantly.*

*Proof.* Firstly, we define the following three events:

$$
\begin{aligned}
\mathcal{E}_{0,m} &= \left\{ \forall t, i, |\hat{\mu}_i(t) - \mu_i| \leq \sqrt{\frac{\log(12mt^2/\delta)}{N_i(t)}} \right\}; \\
\mathcal{E}_{1,m} &= \left\{ \forall i, t, k, |\theta_i^k(t) - \hat{\mu}_i(t)| \leq \sqrt{\frac{C(\delta, q, t) \log(12mt^2 M(\delta, q, t)/\delta)}{N_i(t)}} \right\}; \\
\mathcal{E}_{2,m} &= \left\{ \forall t, |\hat{p}_{\tilde{a}}(t) - p_{\tilde{a}}(t)| \leq \sqrt{\frac{\log(12t^2/\delta)}{M(\delta, q, t)}} \right\}.
\end{aligned}
$$

Here $p_{\tilde{a}}(t)$ denotes the probability that arm $\tilde{a}$'s sample is the largest one in the $k$-th sample set of time step $t$ (i.e., $\Pr[\theta_{\tilde{a}}^k(t) = \max_{i \in [m]} \theta_i^k(t) | \{N_i(t), R_i(t)\}_{i=1}^m]$, also note that it is not conditioned on $\mathcal{E}_{1,m}$). We can see that $\hat{p}_{\tilde{a}}(t)$ is its empirical mean in the $M(\delta, q, t)$ sample sets.

By Chernoff-Hoeffding inequality, $\forall t, i$, $\Pr[|\hat{\mu}_i(t) - \mu_i| > \sqrt{\frac{\log(12mt^2/\delta)}{N_i(t)}}] \leq \frac{\delta}{6mt^2}$, therefore $\Pr[\neg \mathcal{E}_{0,m}] \leq \sum_{i,t} \frac{\delta}{6mt^2} \leq \frac{\delta}{3}$. Similarly, one can also prove that $\Pr[\neg \mathcal{E}_{1,m}] \leq \frac{\delta}{3}$ (note that there are $M(\delta, q, t)$ sample sets at time step $t$, and the variance of $\theta_i^k(t)$ is $\frac{C(\delta, q, t)}{N_i(t)}$) and $\Pr[\neg \mathcal{E}_{2,m}] \leq \frac{\delta}{3}$.

Then we only need to prove that under event $\mathcal{E}_{0,m} \wedge \mathcal{E}_{1,m} \wedge \mathcal{E}_{2,m}$, the two properties in Theorem 1 hold. To simplify the notations, in this proof, we denote $M(t) = M(\delta, q, t)$, $C(t) = C(\delta, q, t)$, $L_1(t) = \log(12mt^2/\delta)$ and $L_2(t) = \log(12mt^2 M(\delta, q, t)/\delta)$.

Consider the two cases in Theorem 1:

i) When $\tilde{a} = a^*$: in this case, we claim that any arm $i$ cannot be pulled if $N_i(t) \geq \frac{16C(t)L_2(t)}{\Delta_{i,m}^2}$, and we will prove this claim by contradiction.

If we choose to pull arm $i \neq a^*$ and $N_i(t) \geq \frac{16C(t)L_2(t)}{\Delta_{i,m}^2}$, then from the description of TS-Verify, there exists $k$ such that $\theta_i^k(t) \geq \theta_{a^*}^k(t)$ and $N_i(t) \leq N_{a^*}(t)$. However, we also have that (in the rest

of this paragraph, $A(t) = \frac{1}{N_i(t)}$ and $B(t) = \frac{1}{N_{a^*}(t)}$)

$$
\begin{aligned}
\theta_i^k(t) &\leq \hat{\mu}_i(t) + \sqrt{A(t)C(t)L_2(t)} && (1) \\
&\leq \mu_i + \sqrt{A(t)L_1(t)} + \sqrt{A(t)C(t)L_2(t)} && (2) \\
&< \mu_i + \frac{\Delta_{i,m}}{2}, && (3)
\end{aligned}
$$

where Ineq. (1) is because of $\mathcal{E}_{1,m}$; Ineq. (2) is because of $\mathcal{E}_{0,m}$; Ineq. (3) is because that $N_i(t) \geq \frac{16C(t)L_2(t)}{\Delta_{i,m}^2}$ and therefore $\sqrt{A(t)C(t)L_2(t)} \leq \frac{\Delta_{i,m}}{4}$ and $\sqrt{A(t)L_1(t)} < \frac{\Delta_{i,m}}{4}$ (since $q \geq \delta$, we must have that $C(t)L_2(t) > L_1(t)$). As for the arm $a^*$, we have that

$$
\theta_{a^*}^k(t) \geq \hat{\mu}_{a^*}(t) - \sqrt{B(t)C(t)L_2(t)} \geq \mu_{a^*} - \sqrt{B(t)L_1(t)} - \sqrt{B(t)C(t)L_2(t)} > \mu_i + \frac{\Delta_{i,m}}{2},
$$

where the last inequality is because that $N_{a^*}(t) \geq N_i(t) \geq \frac{16C(t)L_2(t)}{\Delta_{i,m}^2}$. This contradicts with $\theta_i^k(t) \geq \theta_{a^*}^k(t)$.

Similarly, if $N_{a^*}(t) \geq \frac{16C(t)L_2(t)}{\Delta_{a^*,m}^2}$, then arm $a^*$ cannot be pulled as well. This means that the complexity $Z$ satisfies

$$
Z \leq \sum_{i \in [m]} \frac{16C(Z)L_2(Z)}{\Delta_{i,m}^2} = 16H_m C(Z)L_2(Z).
$$

For $q \leq 0.1$, we have that $\phi^2(3q) = \Theta(\log \frac{1}{3q})$. Then with $C(Z) = \frac{4\log(12mZ^2/\delta)}{\phi^2(3q)}$, $L_2(Z) = \log(12mZ^2 M(Z)/\delta)$ and $M(Z) = \frac{\log(12Z^2/\delta)}{q^2}$, we have that (note that we require $q \geq \delta$)

$$
Z \leq \Theta\left(H_m \frac{\log(mZ) + \log\frac{1}{\delta}}{\log\frac{1}{q}}\left(\log(mZ) + \log\frac{1}{\delta}\right)\right).
$$

Therefore, after some basic calculations, we know that

$$
Z = O\left(H_m \frac{\log(mH_m) + \log\frac{1}{\delta}}{\log\frac{1}{q}}\left(\log(mH_m) + \log\frac{1}{\delta}\right)\right).
$$

ii) When $\tilde{a} \neq a^*$: in this case, we consider the probability that arm $a^*$ has a larger sample than $\tilde{a}$, i.e., $\Pr[\theta_{a^*}^k(t) \geq \theta_{\tilde{a}}^k(t)|\{N_i(t), R_i(t)\}_{i=1}^m]$, and let $A(t) = \frac{1}{N_{\tilde{a}}(t)}$ and $B(t) = \frac{1}{N_{a^*}(t)}$.

Note that $\mu_{a^*} \geq \mu_{\tilde{a}}$, therefore, under event $\mathcal{E}_{0,m}$, we must have that $\hat{\mu}_{a^*}(t) - \hat{\mu}_{\tilde{a}}(t) \geq (\mu_{a^*} - \sqrt{B(t)L_1(t)}) - (\mu_{\tilde{a}} + \sqrt{A(t)L_1(t)}) \geq -\sqrt{A(t)L_1(t)} - \sqrt{B(t)L_1(t)}$. Since $\theta_{a^*}^k(t) - \theta_{\tilde{a}}^k(t)$ is a Gaussian random variable with mean $\hat{\mu}_{a^*}(t) - \hat{\mu}_{\tilde{a}}(t)$ and variance $A(t)C(t) + B(t)C(t)$, we have that (recall that $\Phi(x, \mu, \sigma^2) = \Pr_{X \sim \mathcal{N}(\mu, \sigma^2)}[X \geq x]$):

$$
\begin{aligned}
\Pr[\theta_{a^*}^k(t) \geq \theta_{\tilde{a}}^k(t)] &= \Phi(0, \hat{\mu}_{a^*}(t) - \hat{\mu}_{\tilde{a}}(t), A(t)C(t) + B(t)C(t)) \\
&\geq \Phi(0, -\sqrt{A(t)L_1(t)} - \sqrt{B(t)L_1(t)}, A(t)C(t) + B(t)C(t)) \\
&= \Phi(\sqrt{A(t)L_1(t)} + \sqrt{B(t)L_1(t)}, 0, A(t)C(t) + B(t)C(t)) \\
&= \Phi\left(\sqrt{\frac{L_1(t)}{C(t)}} \cdot \frac{\sqrt{A(t)} + \sqrt{B(t)}}{\sqrt{A(t) + B(t)}}, 0, 1\right) \\
&\geq \Phi\left(2\sqrt{\frac{L_1(t)}{C(t)}}, 0, 1\right) \\
&= 3q, && (4)
\end{aligned}
$$

where Eq. (4) is because that $C(t) = \frac{4L_1(t)}{\phi^2(3q)}$ and $\Phi(\phi(3q), 0, 1) = 3q$ (by definition of $\phi$). This implies that $\Pr[\theta_{\tilde{a}}^k(t) = \max_{i \in [m]} \theta_i^k(t)|\{N_i(t), R_i(t)\}_{i=1}^m] \leq 1 - 3q$.

Note that $p_{\tilde{a}}(t)$ only depends on $\mathcal{E}_{0,m}$ but does not depend on $\mathcal{E}_{1,m}$ and $\mathcal{E}_{2,m}$. Therefore under event $\mathcal{E}_{0,m} \wedge \mathcal{E}_{1,m} \wedge \mathcal{E}_{2,m}$, we still have that $p_{\tilde{a}}(t) \leq 1 - 3q$. Also recall that we set $M(\delta, q, t) = \frac{\log(12t^2/\delta)}{q^2}$, which implies $\sqrt{\frac{\log(12t^2/\delta)}{M(\delta,q,t)}} = q$. Hence, by event $\mathcal{E}_{2,m}$, we know that $\hat{p}_{\tilde{a}}(t) \leq p_{\tilde{a}}(t) + q \leq 1 - 3q + q < 1 - q$, i.e., TS-Verify will not stop in this case. $\qquad\square$

Theorem 1 shows that the complexity upper bound of TS-Verify (as well as the complete pure exploration algorithm constructed by applying the explore-then-verify framework) is asymptotically optimal in the MAB case, i.e., it matches the complexity lower bound $\Omega(H_m \log \frac{1}{\delta})$ (Kalyanakrishnan et al., 2012) when $\delta \to 0$. To the best of our knowledge, this is the first analysis of using a TS-based algorithm to deal with pure exploration problems under the frequentist setting.

**Remark 2.** *Focusing on verification is the mechanism we use to bound the number of pulls on sufficiently learned arms (i.e., arm $i$ with $N_i(t) \geq 16C(t)L_2(t)/\Delta_{i,m}^2$). For a complete pure exploration algorithm, the optimal arm is unknown at the beginning. If it is insufficiently learned, then the variance of its random sample is large, and we cannot ensure that it is optimal in any of the sample sets. In this case, the exploration set may only contain sufficiently learned arms. However, in a verification algorithm, we only focus on the complexity when the target arm is optimal, and we can always put the target arm in the exploration set. This makes sure that there is at least one insufficiently learned arm in the exploration set, i.e., if the target arm is sufficiently learned, then the other arm that has at least one larger random sample than the target arm must be insufficiently learned. Therefore, the number of pulls on sufficiently learned arms in TS-Verify is limited.*

**Remark 3.** *The variance term $C(\delta, q, t)$ in TS-Verify is chosen carefully to both ensure the correctness and reduce the complexity upper bound. Since $C(\delta, q, t)$ is not too low, for any sub-optimal target $\tilde{a}$, there is always some probability that the optimal arm $a^*$ has larger sample than $\tilde{a}$, even if the observations of $a^*$ are bad (e.g., its empirical mean equals to its real expected reward minus the confidence radius). This ensures the correctness of TS-Verify. On the other hand, since $C(\delta, q, t)$ is not too high, the required number of observations on sub-optimal arm $i$ is still $O(\frac{1}{\Delta_i^2} \log \frac{1}{\delta})$.*

## 4.3 TS-VERIFY FOR COMBINATORIAL MULTI-ARMED BANDIT

Then we come to the CMAB case, and we choose $C(\delta, q, t) = \frac{4 \log(24|\mathcal{I}|t^2/\delta)}{\phi^2(3q)}$. For the exploration set $\{S', \tilde{S}\}$, we will choose to pull the arm $i \in (S' \setminus \tilde{S}) \cup (\tilde{S} \setminus S')$ with the smallest number of observations $N_i(t)$.

**Theorem 2.** *For $q \in [\delta, 0.1)$, with probability at least $1-\delta$, we have that: i) if $\tilde{S} = S^*$, TS-Verify will return that super arm $\tilde{S}$ is the optimal super arm with complexity upper bounded by $O(H_{1,c}(\log \frac{1}{\delta} + \log(|\mathcal{I}|H_{1,c}))\frac{\log \frac{1}{\delta}}{\log \frac{1}{q}} + H_{1,c}\frac{\log^2(|\mathcal{I}|H_{1,c})}{\log \frac{1}{q}})$; ii) if $\tilde{S} \neq S^*$, TS-Verify will not stop. Specifically, if we choose $q = \delta$, then the complexity upper bound is $O(H_{1,c} \log \frac{1}{\delta} + H_{1,c} \log^2(|\mathcal{I}|H_{1,c}))$.*

Now we provide a proof sketch for Theorem 2, while deferring its complete proof to Appendix B.

*Proof (sketch).* In the CMAB case, we first define $\mathcal{J} = \{S^* \setminus S, S \setminus S^* : S \in \mathcal{I}\}$ as the exchange sets of any super arm $S \in \mathcal{I}$ and the optimal super arm $S^*$. Then similar as the proof of Theorem 1, we can also define three events $\mathcal{E}_{0,c}, \mathcal{E}_{1,c}, \mathcal{E}_{2,c}$.

$$\mathcal{E}_{0,c} = \left\{ \forall t, U \in \mathcal{J}, |\sum_{i \in U}(\hat{\mu}_i(t) - \mu_i)| \leq \sqrt{\sum_{i \in U}\frac{1}{N_i(t)}\log(24|\mathcal{I}|t^2/\delta)} \right\};$$

$$\mathcal{E}_{1,c} = \left\{ \forall t, k, U \in \mathcal{J}, |\sum_{i \in U}(\theta_i^k(t) - \hat{\mu}_i(t))| \leq \sqrt{C(\delta, q, t)\sum_{i \in U}\frac{1}{N_i(t)}\log(24|\mathcal{I}|t^2 M(\delta, q, t)/\delta)} \right\};$$

$$\mathcal{E}_{2,c} = \left\{ \forall t, |\hat{p}_{\tilde{S}}(t) - p_{\tilde{S}}(t)| \leq \sqrt{\frac{\log(12t^2/\delta)}{M(\delta, q, t)}} \right\}.$$

Here $p_{\tilde{S}}(t)$ denotes the probability that at time step $t$, $\tilde{S}$ is the best super arm under the $k$-th sample set (i.e., $\Pr[\sum_{i \in \tilde{S}} \theta_i^k(t) = \max_{S \in \mathcal{I}} \sum_{i \in S} \theta_i^k(t) | \{N_i(t), R_i(t)\}_{i=1}^m])$. We can see that $\hat{p}_{\tilde{S}}(t)$ is its empirical mean in the $M(\delta, q, t)$ sample sets.

Similarly, we also have that $\Pr[\mathcal{E}_{0,c} \wedge \mathcal{E}_{1,c} \wedge \mathcal{E}_{2,c}] \geq 1 - \delta$, and therefore it is sufficient to prove that under event $\mathcal{E}_{0,c} \wedge \mathcal{E}_{1,c} \wedge \mathcal{E}_{2,c}$, the two properties in Theorem 2 hold. In this proof, we denote $M(t) = M(\delta, q, t)$, $C(t) = C(\delta, q, t)$, $L_1(t) = \log(24|\mathcal{I}|t^2/\delta)$ and $L_2(t) = \log(24|\mathcal{I}|t^2 M(\delta, q, t)/\delta)$.

Consider the two cases in Theorem 2:

i) When $\tilde{S} = S^*$: in this case, we claim that any arm $i$ cannot be pulled if $N_i(t) \geq \frac{16\,\text{width}\,C(t)L_2(t)}{\Delta_{i,c}^2}$, and we will also prove this claim by contradiction.

Here we only consider the case that $i \notin S^*$, the case that $i \in S^*$ is similar. For any arm $i \notin S^*$, if it is pulled at time step $t$ and $N_i(t) \geq \frac{16\,\text{width}\,C(t)L_2(t)}{\Delta_{i,c}^2}$, then from the description of TS-Verify, there exists $S \neq S^*$ and $i \in S \setminus S^*$ such that for some $k$, $\sum_{j \in S} \theta_j^k(t) \geq \sum_{j \in S^*} \theta_j^k(t)$ (which is the same as $\sum_{j \in S \setminus S^*} \theta_j^k(t) \geq \sum_{j \in S^* \setminus S} \theta_j^k(t))$, and $N_j(t) \geq N_i(t)$ for all $j \in (S^* \setminus S) \cup (S \setminus S^*)$. Now let $A(t) = \sum_{j \in S \setminus S^*} \frac{1}{N_j(t)}$ and $B(t) = \sum_{j \in S^* \setminus S} \frac{1}{N_j(t)}$, then the proof is almost the same as Theorem 1 (note that there are at most $\text{width}$ arms in $S \setminus S^*$ or $S^* \setminus S$).

Therefore, we know that the complexity $Z$ satisfies

$$Z \leq \sum_{i \in [m]} \frac{16\,\text{width}\,C(Z)L_2(Z)}{\Delta_{i,c}^2} = 16 H_{1,c} C(Z) L_2(Z),$$

which means that $Z = O(H_{1,c}(\log \frac{1}{\delta} + \log(|\mathcal{I}|H_{1,c}))\frac{\log \frac{1}{\delta}}{\log \frac{1}{q}} + H_{1,c} \frac{\log^2(|\mathcal{I}|H_{1,c})}{\log \frac{1}{q}})$.

ii) When $\tilde{S} \neq S^*$: in this case, we consider the probability that $\sum_{i \in S^* \setminus \tilde{S}} \theta_i^k(t) \geq \sum_{i \in \tilde{S} \setminus S^*} \theta_i^k(t)$ (i.e., super arm $S^*$ is better than $\tilde{S}$ in the $k$-th sample set). Here we denote $A(t) = \sum_{i \in \tilde{S} \setminus S^*} \frac{1}{N_i(t)}$, $B(t) = \sum_{i \in S^* \setminus \tilde{S}} \frac{1}{N_i(t)}$, and then the proof becomes almost the same as Theorem 1. $\qquad \square$

Compared to existing works, the complexity upper bound of TS-Verify (as well as the complete pure exploration algorithm constructed by applying the explore-then-verify framework) is $\text{width}$ lower than the CLUCB policy in (Chen et al., 2014). This is because that we use the Thompson Sampling approach, i.e., for base arms in an exchange set $S^* \setminus S$, the sum of their random samples (in TS-Verify) only has a bias of $\tilde{O}(\sqrt{\sum_{i \in S^* \setminus S} \frac{1}{N_i(t)}})$ towards the sum of their empirical means, while the sum of their upper confidence bounds (in CLUCB) has a bias of $\tilde{O}(\sum_{i \in S^* \setminus S} \sqrt{\frac{1}{N_i(t)}})$. Since the observations of the base arms are independently drawn by the environment, the gap between the sum of their empirical means and the sum of their real means is $\tilde{O}(\sqrt{\sum_{i \in S^* \setminus S} \frac{1}{N_i(t)}})$. Therefore, our TS-Verify algorithm not only has correctness guarantee (with proper $C(\delta, q, t)$), but also achieves a $\text{width}$ lower complexity upper bound. We also conduct some experiments to compare the complexity of TS-Verify with CLUCB, and the experimental results (which are listed in Appendix C) demonstrate the effectiveness of our algorithm.

Though the complexity bound of TS-Verify still has some gap with the optimal one $O(H_{0,c} \log \frac{1}{\delta})$, we emphasize that this is because we do not assume the existence of more powerful offline oracles, and only use a simple offline oracle to solve the combinatorial pure exploration problems. Both the existing optimal policies (Chen et al., 2017; Jourdan et al., 2021) either suffer from an exponential time cost, or require to adopt more powerful offline oracles (please see Appendix D for more details). Therefore, our algorithm is more general, and can be attractive in real applications with large scale (which means a large $\text{width}$) and complex combinatorial structures (which may result in a high implementation cost for the optimal learning algorithms).

**Remark 4.** *If the value $|\mathcal{I}|$ is unknown (which is common in real applications), we can use $2^m$ instead (i.e., $C(\delta, q, t) = \frac{4 \log(24 \cdot 2^m t^2/\delta)}{\phi^2(3q)}$). This only increases the constant term in the complexity upper bound and does not influence the major term $O(H_{1,c} \log \frac{1}{\delta})$.*

ETHICS STATEMENT

Our TS-based algorithm is easy to implement and has low complexity in combinatorial pure exploration. Therefore, many important applications in our life, such as routing systems and online advertising systems, can benefit from this new approach, i.e., the systems can find out their optimal actions with much lower cost than before. Besides, the idea of using TS in combinatorial pure exploration may push the development of algorithms in other pure exploration problems. On the other hand, since this paper only contains pure theoretical results, its negative impact on society is not obvious.

REPRODUCIBILITY STATEMENT

To make our theoretical results reproducible, we include the complete proof of Theorem 1 in Section 4.2 and the complete proof of Theorem 2 in Appendix B. Besides, we attach the detailed descriptions of the general explore-then-verify framework (Chen et al., 2017) in Appendix A. This can help to understand Fact 1, i.e., how to construct a complete pure exploration algorithm with complexity bound asymptotically the same as a verification algorithm.

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

APPENDIX

## A  GENERAL EXPLORE-THEN-VERIFY FRAMEWORK

Algorithm 2 outlines the general explore-then-verify framework (Chen et al., 2017).

In this framework, there are several independent procedures, and the $k$-th procedure is active at time step $t$ only if $t \mod 2^k = 0$. Each procedure $k$ contains two stages: an exploration stage and a verification stage. The exploration stage of procedure $k$ is a complete pure exploration algorithm, i.e., it outputs the optimal arm with probability at least $1 - \delta_0$ (here we choose $\delta_0 = 0.1$). After the exploration stage of procedure $k$ outputs an arm, the verification stage of procedure $k$ then tries to verify that the output arm is the optimal arm with confidence $1 - \frac{\delta}{2^k}$. Once a verification stage (of any procedure) outputs that its input arm is the optimal arm, then all the procedures terminate, and the explore-then-verify framework outputs that arm as the result of pure exploration. We can see that all these steps form a complete pure exploration algorithm with error constraint $\delta$ (which is different with the complete pure exploration algorithm in the exploration stage with error constraint $\delta_0$).

---

**Algorithm 2** Explore-then-verify framework

1: **Input:** Error constraint $\delta$.
2: **Init:**  For any $k \geq 1$, procedure $k$ contains two stages. In the first stage, it explores with constant error probability $\delta_0 = 0.1$; in the second stage, it tries to verify that the output arm of the exploration stage is the optimal arm with error probability $\frac{\delta}{2^k}$.
3: At time step $t$, if $t \mod 2^k = 0$, then procedure $k$ is active (i.e., it can choose an arm to pull).
4: If in any procedure $k$, the output arm of its exploration stage gets through the verification stage, then output that arm and terminate the algorithm (as well as any other procedures).

---

## B  PROOF OF THEOREM 2

Recall that $\mathcal{J} = \{S^* \setminus S, S \setminus S^* : S \in \mathcal{I}\}$, and the three events $\mathcal{E}_{0,c}, \mathcal{E}_{1,c}, \mathcal{E}_{2,c}$ are defined as following.

$$
\mathcal{E}_{0,c} = \left\{ \forall t, U \in \mathcal{J}, |\sum_{i \in U} (\hat{\mu}_i(t) - \mu_i)| \leq \sqrt{\sum_{i \in U} \frac{1}{N_i(t)} \log(24|\mathcal{I}|t^2/\delta)} \right\};
$$

$$
\mathcal{E}_{1,c} = \left\{ \forall t, k, U \in \mathcal{J}, |\sum_{i \in U} (\theta_i^k(t) - \hat{\mu}_i(t))| \leq \sqrt{C(\delta, q, t) \sum_{i \in U} \frac{1}{N_i(t)} \log(24|\mathcal{I}|t^2 M(\delta, q, t)/\delta)} \right\};
$$

$$
\mathcal{E}_{2,c} = \left\{ \forall t, |\hat{p}_{\tilde{S}}(t) - p_{\tilde{S}}(t)| \leq \sqrt{\frac{\log(12t^2/\delta)}{M(\delta, q, t)}} \right\}.
$$

Note that the random variable $(\hat{\mu}_i(t) - \mu_i)$ is zero-mean and $\frac{1}{N_i(t)}$ sub-gaussian, and for different $i$, the random variables $(\hat{\mu}_i(t) - \mu_i)$'s are independent. Therefore, $\sum_{i \in U} (\hat{\mu}_i(t) - \mu_i)$ is zero-mean and $\sum_{i \in U} \frac{1}{N_i(t)}$ sub-Gaussian. Then by concentration inequality,

$$
\Pr\left[ |\sum_{i \in U} (\hat{\mu}_i(t) - \mu_i)| > \sqrt{\sum_{i \in U} \frac{1}{N_i(t)} \log(24|\mathcal{I}|t^2/\delta)} \right] \leq 2 \exp(-\log(24|\mathcal{I}|t^2/\delta)) = \frac{\delta}{12|\mathcal{I}|t^2}.
$$

This implies that

$$
\Pr[\neg \mathcal{E}_{0,c}] \leq \sum_{U,t} \frac{\delta}{12|\mathcal{I}|t^2} \leq \sum_t \frac{\delta}{6t^2} \leq \frac{\delta}{3},
$$

where the second inequality is because that $|\mathcal{J}| \leq 2|\mathcal{I}|$, and the third inequality is because that $\sum_t \frac{1}{t^2} \leq 2$.

Similarly, the random variable $(\theta_i^k(t) - \hat{\mu}_i(t))$ is a zero-mean Gaussian random variable with variance $\frac{C(\delta,q,t)}{N_i(t)}$, and for different $i$, the random variables $(\theta_i^k(t) - \hat{\mu}_i(t))$'s are also independent. Then by concentration inequality,

$$\Pr\left[|\sum_{i \in U}(\theta_i^k(t) - \hat{\mu}_i(t))| > \sqrt{C(\delta,q,t)\sum_{i \in U}\frac{1}{N_i(t)}\log(24|\mathcal{I}|t^2M(\delta,q,t)/\delta)}\right]$$
$$\leq \quad 2\exp(-\log(24|\mathcal{I}|t^2M(\delta,q,t)/\delta))$$
$$= \quad \frac{\delta}{12|\mathcal{I}|t^2M(\delta,q,t)}.$$

This implies that

$$\Pr[\neg\mathcal{E}_{1,c}] \leq \sum_{U,t,k}\frac{\delta}{12|\mathcal{I}|t^2M(\delta,q,t)} \leq \sum_{U,t}\frac{\delta}{12|\mathcal{I}|t^2} \leq \sum_t\frac{\delta}{6t^2} \leq \frac{\delta}{3},$$

where the second inequality is because that there are totally $M(\delta,q,t)$ sample sets at time step $t$.

Let $Y_k(t) = \mathbb{I}[S^k(t) = \tilde{S}]$, then we know that $\{Y_k(t)\}_{k=1}^{M(\delta,q,t)}$ are $M(\delta,q,t)$ i.i.d.s with real mean $p_{\tilde{S}}(t)$ and empirical mean $\hat{p}_{\tilde{S}}(t)$. Therefore

$$\Pr\left[|\hat{p}_{\tilde{S}}(t) - p_{\tilde{S}}(t)| \leq \sqrt{\frac{\log(12t^2/\delta)}{M(\delta,q,t)}}\right] \leq 2\exp(-\log(12t^2/\delta)) = \frac{\delta}{6t^2},$$

which implies that

$$\Pr[\neg\mathcal{E}_{2,c}] \leq \sum_t\frac{\delta}{6t^2} \leq \frac{\delta}{3}.$$

Now it is sufficient to prove that under event $\mathcal{E}_{0,c} \wedge \mathcal{E}_{1,c} \wedge \mathcal{E}_{2,c}$, the two properties in Theorem 2 hold. In this proof, we denote $M(t) = M(\delta,q,t)$, $C(t) = C(\delta,q,t)$, $L_1(t) = \log(24|\mathcal{I}|t^2/\delta)$ and $L_2(t) = \log(24|\mathcal{I}|t^2M(\delta,q,t)/\delta)$.

Consider the two cases in Theorem 2:

i) When $\tilde{S} = S^*$: in this case, we claim that any arm $i$ cannot be pulled if $N_i(t) \geq \frac{16\,\text{width}\,C(t)L_2(t)}{\Delta_{i,c}^2}$, and we will prove this claim by contradiction.

For any arm $i \notin S^*$ (the proof for arms $i \in S^*$ is similar), if it is pulled at time step $t$ and $N_i(t) \geq \frac{16\,\text{width}\,C(t)L_2(t)}{\Delta_{i,c}^2}$, then from the description of TS-Verify, there exists $S \neq S^*$ and $i \in S \setminus S^*$ such that for some $k$, $\sum_{j \in S}\theta_j^k(t) \geq \sum_{j \in S^*}\theta_j^k(t)$ (which is the same as $\sum_{j \in S \setminus S^*}\theta_j^k(t) \geq \sum_{j \in S^* \setminus S}\theta_j^k(t)$), and $N_j(t) \geq N_i(t)$ for all $j \in (S^* \setminus S) \cup (S \setminus S^*)$. Now let $A(t) = \sum_{j \in S \setminus S^*}\frac{1}{N_j(t)}$ and $B(t) = \sum_{j \in S^* \setminus S}\frac{1}{N_j(t)}$, we have that

$$\sum_{j \in S \setminus S^*}\theta_j^k(t) \leq \sum_{j \in S \setminus S^*}\hat{\mu}_j(t) + \sqrt{A(t)C(t)L_2(t)} \tag{5}$$
$$\leq \sum_{j \in S \setminus S^*}\mu_j + \sqrt{A(t)L_1(t)} + \sqrt{A(t)C(t)L_2(t)} \tag{6}$$
$$< \sum_{j \in S \setminus S^*}\mu_j + \frac{\Delta_{i,m}}{2} \tag{7}$$
$$\leq \sum_{j \in S \setminus S^*}\mu_j + \frac{\Delta_{S,m}}{2},$$

where Ineq. (5) is because of $\mathcal{E}_{1,c}$; Ineq. (6) is because of $\mathcal{E}_{0,c}$; Ineq. (7) is because that $\forall j \in S \setminus S^*, N_j(t) \geq N_i(t) \geq \frac{16\,\text{width}\,C(t)L_2(t)}{\Delta_{i,m}^2}$ and therefore $\sqrt{A(t)C(t)L_2(t)} \leq \frac{\Delta_{i,m}}{4}$ and $\sqrt{A(t)L_1(t)} < \frac{\Delta_{i,m}}{4}$ (since $q \geq \delta$, we must have that $C(t)L_2(t) > L_1(t)$). Similarly, for arms $j \in S^* \setminus S$, we have that

$$
\begin{aligned}
\sum_{j \in S^* \setminus S} \theta_j^k(t) &\geq \sum_{j \in S^* \setminus S} \hat{\mu}_j(t) - \sqrt{B(t)C(t)L_2(t)} \\
&\geq \sum_{j \in S^* \setminus S} \mu_j - \sqrt{B(t)L_1(t)} - \sqrt{B(t)C(t)L_2(t)} \\
&> \sum_{j \in S^* \setminus S} \mu_j - \frac{\Delta_{i,m}}{2} \\
&\geq \sum_{j \in S \setminus S^*} \mu_j + \frac{\Delta_{S,m}}{2},
\end{aligned}
\tag{8}
$$

where Ineq. (8) is because that $\forall j \in S^* \setminus S, N_j(t) \geq N_i(t) \geq \frac{16\,\text{width}\,C(t)L_2(t)}{\Delta_{i,m}^2}$. This contradicts with $\sum_{j \in S \setminus S^*} \theta_j^k(t) \geq \sum_{j \in S^* \setminus S} \theta_j^k(t)$.

Therefore, we know that the complexity $Z$ satisfies

$$
Z \leq \sum_{i \in [m]} \frac{16\,\text{width}\,C(Z)L_2(Z)}{\Delta_{i,c}^2} = 16H_{1,c}C(Z)L_2(Z).
$$

With $C(Z) = \frac{4\log(24|\mathcal{I}|Z^2/\delta)}{\phi^2(3q)}$, $\phi^2(3q) = \Theta(\log \frac{1}{3q})$, $L_2(Z) = \log(24|\mathcal{I}|Z^2 M(Z)/\delta)$ and $M(Z) = \frac{\log(12Z^2/\delta)}{q^2}$, we have that (note that we require $q \geq \delta$)

$$
Z \leq \Theta\left(H_{1,c}\frac{\log(|\mathcal{I}|Z) + \log\frac{1}{\delta}}{\log\frac{1}{q}}\left(\log(|\mathcal{I}|Z) + \log\frac{1}{\delta}\right)\right).
\tag{9}
$$

Then we can use the following lemma to find an upper bound for complexity $Z$.

**Lemma 1.** *Given $K$ functions $f_1(x), \cdots, f_K(x)$ and $K$ positive values $X_1, \cdots, X_K$, if $\forall x \geq X_k, Kf_k(x) < x$ holds for all $1 \leq k \leq K$, then for any $x \geq \sum_k X_k, \sum_k f_k(x) < x$.*

*Proof.* Since $X_1, \cdots, X_K$ are positive values, for any $x \geq \sum_k X_k$, we must have that $x \geq X_k$. Therefore $Kf_k(x) < x$, which implies that

$$
\sum_k Kf_k(x) < \sum_k x.
$$

This is the same as $\sum_k f_k(x) < x$. $\qquad\qquad\square$

To apply Lemma 1, we set $f_1(Z) = H_{1,c}\frac{\log^2(|\mathcal{I}|Z)}{\log\frac{1}{q}}$, $f_2(Z) = H_{1,c}\frac{\log(|\mathcal{I}|Z)\log\frac{1}{\delta}}{\log\frac{1}{q}}$, and $f_3(Z) = H_{1,c}\frac{\log^2\frac{1}{\delta}}{\log\frac{1}{q}}$. After some basic calculations, we get that $X_3 = \Theta(H_{1,c}\frac{\log^2\frac{1}{\delta}}{\log\frac{1}{q}})$, $X_2 = \Theta(H_{1,c}\frac{\log(|\mathcal{I}|H_{1,c})\log\frac{1}{\delta}}{\log\frac{1}{q}} + X_3)$ and $X_1 = \Theta(H_{1,c}\frac{\log^2(|\mathcal{I}|H_{1,c})}{\log\frac{1}{q}})$.

Then we know that for $Z \geq \Theta(H_{1,c}(\log\frac{1}{\delta} + \log(|\mathcal{I}|H_{1,c}))\frac{\log\frac{1}{\delta}}{\log\frac{1}{q}} + H_{1,c}\frac{\log^2(|\mathcal{I}|H_{1,c})}{\log\frac{1}{q}})$, $f_1(Z) + f_2(Z) + f_3(Z) < Z$ (by Lemma 1). This contradicts with Eq. (9). Therefore, we know that $Z = O(H_{1,c}(\log\frac{1}{\delta} + \log(|\mathcal{I}|H_{1,c}))\frac{\log\frac{1}{\delta}}{\log\frac{1}{q}} + H_{1,c}\frac{\log^2(|\mathcal{I}|H_{1,c})}{\log\frac{1}{q}})$.

ii) When $\tilde{S} \neq S^*$: in this case, we consider the probability that $\sum_{i \in S^* \setminus \tilde{S}} \theta_i^k(t) \geq \sum_{i \in \tilde{S} \setminus S^*} \theta_i^k(t)$, i.e., super arm $S^*$ is better than $\tilde{S}$ in the $k$-th sample set. Here we denote $A(t) = \sum_{i \in \tilde{S} \setminus S^*} \frac{1}{N_i(t)}$, $B(t) = \sum_{i \in S^* \setminus \tilde{S}} \frac{1}{N_i(t)}$.

Note that $\sum_{i \in S^* \setminus \tilde{S}} \mu_i \geq \sum_{i \in \tilde{S} \setminus S^*} \mu_i$, therefore, under event $\mathcal{E}_{0,c}$, we must have that $\sum_{i \in S^* \setminus \tilde{S}} \hat{\mu}_i(t) - \sum_{i \in \tilde{S} \setminus S^*} \hat{\mu}_i(t) \geq (\sum_{i \in S^* \setminus \tilde{S}} \mu_i - \sqrt{B(t)L_1(t)}) - (\sum_{i \in \tilde{S} \setminus S^*} \mu_i + \sqrt{A(t)L_1(t)}) \geq -\sqrt{A(t)L_1(t)} - \sqrt{B(t)L_1(t)}$. Since $\sum_{i \in S^* \setminus \tilde{S}} \theta_i^k(t) - \sum_{i \in \tilde{S} \setminus S^*} \theta_i^k(t)$ is a Gaussian random variable with mean $\sum_{i \in S^* \setminus \tilde{S}} \hat{\mu}_i(t) - \sum_{i \in \tilde{S} \setminus S^*} \hat{\mu}_i(t)$ and variance $A(t)C(t) + B(t)C(t)$, we have that (recall that $\Phi(x, \mu, \sigma^2) = \Pr_{X \sim \mathcal{N}(\mu, \sigma^2)}[X \geq x]$):

$$
\begin{aligned}
\Pr\left[ \sum_{i \in S^* \setminus \tilde{S}} \theta_i^k(t) \geq \sum_{i \in \tilde{S} \setminus S^*} \theta_i^k(t) \right] &= \Phi\left(0, \sum_{i \in S^* \setminus \tilde{S}} \hat{\mu}_i(t) - \sum_{i \in \tilde{S} \setminus S^*} \hat{\mu}_i(t), A(t)C(t) + B(t)C(t) \right) \\
&\geq \Phi(0, -\sqrt{A(t)L_1(t)} - \sqrt{B(t)L_1(t)}, A(t)C(t) + B(t)C(t)) \\
&= \Phi(\sqrt{A(t)L_1(t)} + \sqrt{B(t)L_1(t)}, 0, A(t)C(t) + B(t)C(t)) \\
&= \Phi\left( \sqrt{\frac{L_1(t)}{C(t)}} \cdot \frac{\sqrt{A(t)} + \sqrt{B(t)}}{\sqrt{A(t) + B(t)}}, 0, 1 \right) \\
&\geq \Phi\left( 2\sqrt{\frac{L_1(t)}{C(t)}}, 0, 1 \right) \\
&= 3q, \quad\quad\quad\quad\quad\quad\quad\quad\quad\quad\quad\quad\quad (10)
\end{aligned}
$$

where Eq. (10) is because that $C(t) = \frac{4L_1(t)}{\phi^2(3q)}$ and $\Phi(\phi(3q), 0, 1) = 3q$ (by definition of $\phi$). This implies that $\Pr[\sum_{i \in \tilde{S}} \theta_i^k(t) = \max_{S \in \mathcal{I}} \sum_{i \in S} \theta_i^k(t) | \{N_i(t), R_i(t)\}_{i=1}^m] \leq 1 - 3q$.

Note that $p_{\tilde{S}}(t)$ only depends on $\mathcal{E}_{0,c}$ but does not depend on $\mathcal{E}_{1,c}$ and $\mathcal{E}_{2,c}$. Therefore under event $\mathcal{E}_{0,c} \wedge \mathcal{E}_{1,c} \wedge \mathcal{E}_{2,c}$, we still have that $p_{\tilde{S}}(t) \leq 1 - 3q$. Also recall that we set $M(\delta, q, t) = \frac{\log(12t^2/\delta)}{q^2}$, which implies $\sqrt{\frac{\log(12t^2/\delta)}{M(\delta,q,t)}} = q$. Therefore, by event $\mathcal{E}_{2,c}$, we know that $\hat{p}_{\tilde{S}}(t) \leq p_{\tilde{S}}(t) + q \leq 1 - 3q + q < 1 - q$, i.e., TS-Verify will not stop in this case.

## C  EXPERIMENTS

Here we conduct a simple experiment to compare the complexity of our algorithm with CLUCB, and to demonstrate the effectiveness of our algorithm. In this experiment, we consider the following combinatorial pure exploration problem instance with parameter $n \in \mathbb{N}^+$.

**Problem 1.** *For fixed value $n$, there are totally $2n$ base arms. For the first $n$ base arms, their expected rewards equal to 0.1, and for the last $n$ base arms, their expected rewards equal to 0.9. There are only two super arms $S_1, S_2$. $S_1$ contains the first $n$ base arms and $S_2$ contains the last $n$ base arms.*

Since there are only two super arms, our TS-Verify can also be used to do pure exploration, i.e., at time step $t$, if $S_1$ is the optimal super arm in at least $(1 - q)M(\delta, q, t)$ sample sets, then we output that $S_1$ is optimal; if $S_2$ is the optimal super arm in at least $(1 - q)M(\delta, q, t)$ sample sets, then we output that $S_2$ is optimal; otherwise we pull all the arms in a round robin method. In this way, we can compare TS-Verify with CLUCB directly. As for the input $q$ in TS-Verify, we always choose $q = \delta$.

We first fix $\delta = 0.1$, and compare the complexity of TS-Verify with CLUCB under different $n$. The results (average complexities and their standard deviations) take an average over 20 independent runs, and are shown in Fig. 1(a). We can see that when $n$ increases, the complexity of TS-Verify does not increase a lot, while the complexity of CLUCB increases linearly. This accords with our analysis, since $H_{1,c}(n) = 2n \cdot \frac{2n}{(0.8n)^2} = 6.25$ is a constant but $H_{2,c}(n) = 2n \cdot \frac{(2n)^2}{(0.8n)^2} = 12.5n$ is linear with $n$ (here $H_{1,c}(n), H_{2,c}(n)$ are the values of $H_{1,c}, H_{2,c}$ under the problem instance with parameter $n$, respectively).

Then we fix $n = 10$, and compare the complexity of TS-Verify with CLUCB under different $\delta$. The results (average complexities and their standard deviations) take an average over 20 independent runs, and are shown in Fig. 1(b). We can see that when $\delta$ is large, the complexity of TS-Verify

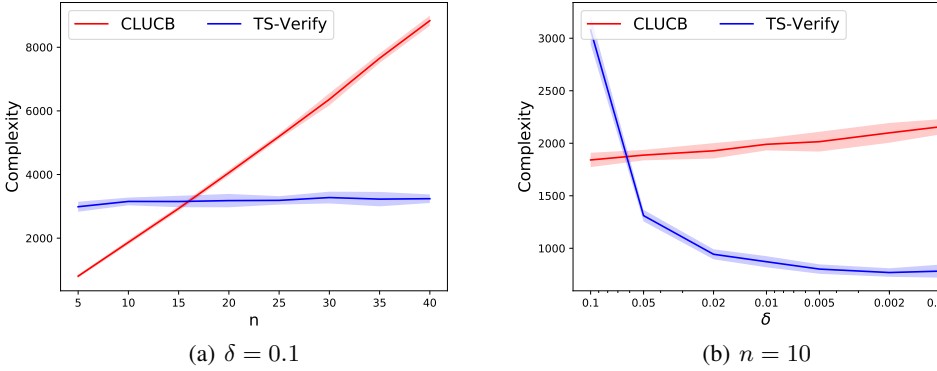

(a) $\delta = 0.1$        (b) $n = 10$

Figure 1: Comparison of TS-Verify and CLUCB.

decreases as $\delta$ decreases, and when $\delta$ is small, the complexity of TS-Verify increases as $\delta$ decreases. Moreover, the complexity of TS-Verify always increases slower than CLUCB (when $\delta$ decreases). This also accords with our analysis. Note that there is a term $O(H_{1,c} \frac{\log^2(|\mathcal{I}|H_{1,c})}{\log \frac{1}{q}})$ in the complexity bound in Theorem 2. Since we choose $q = \delta$, this term decreases as $\delta$ decreases. When $\delta = 0.1$, this term is very large and becomes the majority term in complexity, and therefore the complexity decreases when we decrease $\delta$ from 0.1 to 0.002. When $\delta = 0.002$, the term $O(H_{1,c} \log \frac{1}{\delta})$ becomes the majority term in complexity, therefore the complexity increases when we decrease $\delta$ from 0.002 to 0.001.

All the above results indicate that our TS-Verify algorithm outperforms CLUCB.

## D   DISCUSSIONS ABOUT THE OFFLINE ORACLES IN OPTIMAL ALGORITHMS FOR COMBINATORIAL PURE EXPLORATION

Most existing works on combinatorial pure exploration (e.g., (Chen et al., 2014; 2017; Jourdan et al., 2021)) need to use the basic offline oracle, i.e., given a set of parameters $[\theta_1, \cdots, \theta_m]$, it can output $\arg\max_{S \in \mathcal{I}} \sum_{i \in S} \theta_i$, the best super arm under this parameter set. Similar with (Chen et al., 2014), our TS-Verify algorithm also only needs to use this offline oracle, and it is efficient (i.e., its time complexity is polynomial with $m$) as long as this basic offline oracle is efficient.

However, the algorithms in (Chen et al., 2017; Jourdan et al., 2021) require efficient and more powerful oracles. Otherwise the time complexity of their algorithms becomes exponential with $m$. In (Chen et al., 2017), the NaiveGapElim algorithm needs to record all the super arms in $\mathcal{I}$. This means that its time complexity can be exponential with $m$. As for the EfficientGapElim algorithm, it needs to solve three offline problems: OPT, Check and Unique, and directly solving these three problems leads to an exponential time cost as well. The authors apply an approximate version of Pareto curves to solve these problems, but this approach is only efficient under some special combinatorial structures. Therefore, for the most general combinatorial pure exploration setting, EfficientGapElim either suffers from an exponential time complexity, or requires more powerful offline oracles to solve the above three problems efficiently.

In (Jourdan et al., 2021), the best-response oracle used by the $\lambda$-player has time cost scaled with $|N(S_t)|$, where $S_t$ is a super arm, $N(S_t)$ represents the set of super arms whose cells' boundaries intersect the boundary of the cell of $S_t$, and the cell of a super arm $S_t$ is defined as all the possible parameter sets $[\theta_1, \cdots, \theta_m]$ in which $S_t$ is the optimal super arm. Though the size of $N(S_t)$ is always smaller than $\mathcal{I}$, it can still be exponential with $m$. Therefore, the CombGame algorithm in (Jourdan et al., 2021) either suffers from an exponential time complexity, or requires more powerful offline oracles to substitute the origin best-response oracle used by the $\lambda$-player.

In fact, existing optimal learning policies need to use the more powerful offline oracles to efficiently find the base arm that needs exploration the most. Here we say base arm $i$ needs exploration the most at time $t$ if $\alpha N_i^* \log \frac{1}{\delta} - N_i(t)$ is the largest one among the set of base arms, where $\alpha$ is some universal constant, $N_i^*$ is the value of $N_i$ in the optimal solution $H_{0,c}$. From this definition, we can see that if we always pull the base arm that needs exploration the most, then the frequency of pulling each base arm converges to the optimal solution of $H_{0,c}$, and this achieves the optimal complexity upper bound. However, the basic offline oracle in TS-Verify can only find several insufficiently learned super arms (i.e., the super arms that have been optimal in at least one sample set), and cannot tell us which base arm (in these insufficiently learned super arms) needs exploration the most. Therefore, if we want to use a more powerful offline oracle in TS-verify to achieve an optimal complexity upper bound, then this offline oracle needs to tell us which base arm in these insufficiently learned super arms needs exploration the most. If this powerful offline oracle can always output the base arm that needs exploration the most, then the complexity upper bound of TS-Verify is reduced to $O(H_{0,c} \log \frac{1}{\delta})$ and becomes optimal. How to design such offline oracles (to look for the base arm that needs exploration the most in our TS-Verify framework) and how to implement them efficiently is one of our future research topics.

