# OpenReview forum: "Thompson Sampling for (Combinatorial) Pure Exploration"
_ICLR.cc/2022/Conference — ICLR 2022 Submitted_

### Official Review · Reviewer_6x9v · 2021-11-01

**Correctness:** 3
**Technical Novelty And Significance:** 3
**Empirical Novelty And Significance:** Not applicable
**Recommendation:** 5
**Confidence:** 4

**Main Review:**

Strong points
a) It is the first TS-based framework to solve the verification problem in pure exploration.

b) The verification framework achieves lower complexity.

Weak points
a) The paper mainly emphasizes the lower complexity of TS-verify compared with CLUCB. But CLUCB is a complete pure exploration framework while TS-verify can only determine whether the input is the optimal one. So, I think these two results are not directly comparable.

b) The paper only proposes the TS-based verification framework but not the complete pure exploration algorithm. The authors claim that this verification framework combined with a complete pure exploration algorithm, such as CLUCB, can be used to find the best (super) arm. First, for completeness, I recommend authors include the pure exploration algorithm combined with TS-verify in the paper. Second, the authors should provide the complexity of this full framework. In this framework, does the complexity of CLUCB dominate the full complexity? If so, what is the advantage of the proposed algorithm to solve the pure exploration problem?

c) Since it is the first TS-based algorithm to solve the pure exploration problem, it would be better to provide experimental verifications and comparisons to support the claims.

d) It is a bit strange that the algorithm can go on forever in some cases.

------
Thanks to the authors for the comments. I'm still not convinced by the setting and results. So I would rather keep my score.


**Summary Of The Paper:**

This paper designs the first TS-based algorithm to solve the pure exploration problem for both multi-armed bandit (MAB) and combinatorial MAB (CMAB) problems. It mainly focuses on the verification problem where a target arm is given and the algorithm needs to determine whether it is the optimal one.

Benefiting from independent samples of the TS algorithm, the TS-based verification framework achieves lower complexity than the UCB-based one.


**Summary Of The Review:**

I think the formulated problem is a bit strange which is not a standard pure exploration problem. And the algorithm will not stop if the input arm is not optimal. So I would suggest authors give a complete pure exploration algorithm together with the complexity analysis.

---

### Official Review · Reviewer_YozJ · 2021-11-02

**Correctness:** 3
**Technical Novelty And Significance:** 3
**Empirical Novelty And Significance:** 2
**Recommendation:** 5
**Confidence:** 2

**Main Review:**

For pure exploration, existing works based on UCB approaches, They use the sum of upper confidence bounds within arm set S to judge whether S is optimal. This sum can be much larger than the exact upper confidence bound of S,  so the UCB approach requires much higher complexity than necessary. To solve this problem, author introduce the Thompson sampling to it.  Compared to these results, the TS-Verify policy uses a totally different approach, and can achieve an asymptotically optimal complexity upper bound as well. TS-based algorithm achieves a better complexity bound than CLUCB in the most general setting. However, since they do not assume the existence of a more powerful offline oracle, so cannot achieve the optimal complexity upper bound. The contributions seems not enough. The paper structure and writing is good.

**Summary Of The Paper:**

Pure exploration plays an important role in online learning. Existing work mainly focuses on the UCB faces some challenges when looking for the best arm set under some specific combinatorial structures, so authors explore the idea of Thompson Sampling (TS) that uses independent random samples instead of the upper confidence bounds to make decisions, and design the first TS-based algorithm framework TS-Verify. In TS-Verify, the sum of independent random samples within arm set S will not exceed the exact upper confidence bound of S with high probability. As for pure exploration of classic multi-armed bandit, they show that TS-Verify achieves an asymptotically optimal complexity upper bound.
The TS-based verification framework TS- Verify takes a target (super) arm as input, and aims to verify that this target (super) arm is optimal with error constraint . At each time step, TS-Verify first draws independent random samples  for all the (base) arms. Then it tries to find out the M best (super) arms under sample sets.  If in most of these sample sets, the target (super) arm is the best one, then the algorithm will output that the target (super) arm is indeed optimal. Otherwise, the algorithm will choose to pull a (base) arm from the exploration set, which contains the target (super) arm and all the (super) arms that have been optimal in at least one sample set at this time step.


**Summary Of The Review:**

The algorithm works well, but the contribution may not be enough.

---

### Official Review · Reviewer_TkoG · 2021-11-02

**Correctness:** 4
**Technical Novelty And Significance:** 2
**Empirical Novelty And Significance:** Not applicable
**Recommendation:** 6
**Confidence:** 4

**Main Review:**

Strengths:
* The analysis of TS-Verify is correct in my view as I don’t see any errors. The techniques used are well-known and a similar analysis works for verification in both vanilla multi-armed bandit as well as the combinatorial bandit problem.

Weaknesses:
* The sample complexity obtained is in terms of $H_{1,c}$ which is larger than the lower bound that is in terms of $H_{0,c}$. It is said that this is possibly due to not having a powerful offline oracle that is assumed in existing works of [Chen et al 2017] and [Jourdan et al 2021]. But it is not clear to me how the oracles used in those works are useful for the problem considered here. In [Chen et al 2017] the oracle is used to reduce the computational complexity (as opposed to sample complexity) of their algorithm when the set $\mathcal{I}$ of super-arms has a size exponential in the number of arms; the assumed oracle allows them to identify the best super-arm in $\mathcal{I}$ given a set of means using polynomial computation in the number of arms. On the other hand, in [Jourdan et al 2021] the oracle is used to solve a max-min zero-sum game between the learner and the environment. There the oracle is assumed to provide the “best-response” allocation for any choice of arm means given by the environment. Other oracles considered in the literature are those used in tracking algorithms such as Track-and-Stop by [Garivier and Kaufmann 2016, http://proceedings.mlr.press/v49/garivier16a.pdf ] which find the optimal arm allocation for identifying the best arm if a given vector is indeed the true set of arm means. More clarification on which oracle and how it can be used for verification would be helpful.
* The introduction describes at a high level why the analysis of TS is more involved than UCB-type analysis. However, the problem being solved using TS in existing literature is regret minimization. Indeed in works such as [Agrawal and Goyal 2013] and [Kaufmann et al 2012] the analysis of TS required the introduction of technically novel arguments-- Lemma 1 in [Agrawal and Goyal 2013] relates the conditional probability of sampling a suboptimal arm as being directly proportional to the conditional probability of sampling the optimal arm with the coefficient of proportionality decreasing, while Proposition 1 of [Kaufmann et al 2012] shows that the probability of sampling the optimal arm few times must be low by explicitly reasoning about the random number of sub-optimal arm samples taken between successive samples of the optimal arm. On the other hand, the TS analysis in the present paper uses standard concentration arguments, possibly because it is only concerned with verification and not exploration. In the regret minimization case, both objectives need to be simultaneously balanced.
* At a high level, the motivation as to why a TS approach should be used for the verification subroutine is unexplained. Is it easier to implement than competing approaches, or applicable in more general situations? It is known that TS has shown good empirical performance in minimizing regret [Chapelle and Li 2011, https://papers.nips.cc/paper/2011/hash/e53a0a2978c28872a4505bdb51db06dc-Abstract.html ], however no simulations or real-data experiments were shown to demonstrate a similar benefit in the verification problem by the present paper.

Minor questions / comments:
* In TS-Verify, why was the Gaussian distribution used in Line 5? I imagine the analysis would hold with a distribution such as Uniform centered at the required non-zero mean with appropriately chosen width. If so, it could be mentioned that the Gaussian was chosen for convenience.
* In the introduction it is mentioned that Bayesian regret loses many good properties in the frequentist setting. The text after that mentioning that the parameters of the game are chosen in i.i.d. Manner were confusing to me. Could you elaborate more on this, possibly referring to Section 3.1 in [Russo and Van Roy 2013, https://arxiv.org/abs/1301.2609 ].
* In the analysis of Section 4.2, I felt that using A(t) and B(t) was probably not helpful, using $1/N_i(t)$ would be clearer. When stating that $N_{a*}(t) \geq N_i(t)$, it could be helpful to remind that this is because of the strategy to pull the arm with a lesser number of observations.
* It would be easier to use set symmetric difference notation to describe sets like $\mathcal{J}$.



**Summary Of The Paper:**

The paper provides a Thompson Sampling (TS) algorithm for verifying whether a given arm is indeed the best arm (or best super-arm in a combinatorial bandit) with fixed confidence and its theoretical analysis. The proposed algorithm TS-Verify can be used as a subroutine within the explore-then-verify framework of [Karnin 2016, Chen et al 2017] which identifies the best arm (or super-arm) with fixed confidence. The theoretical analysis of the algorithm shows it uses a factor of width fewer samples for verification in combinatorial bandits compared to existing CLUCB algorithm [Chen et al 2014]. The key step in the analysis is defining three events that occur with high probability. $E_{0,m}$ is a concentration event due to independent samples from each arm distribution, $E_{1,m}$ is a single sample Gaussian tail and $E_{2,m}$ is a concentration event for independent Bernoulli samples. These events specify random events for all times, and each time step $t$ defines an independent probability measure. The analysis for the combinatorial bandit follows a similar path, and at a high level, the improved sample complexity is obtained due to the concentration of event $E_{0,c}$ in appendix.

**Summary Of The Review:**

Pure exploration in bandits is a well-studied problem and Thompson Sampling is a well-known meta-algorithm. The paper provides a TS-based algorithm for best arm verification in MAB (or super-arm verification in combinatorial MAB). While the analysis is correct, more discussion about why the simpler (in comparison to [Agrawal and Goyal 2013] or [Kaufmann et al 2012]) analysis works in this problem would be helpful. Motivation for why TS should be used for verification could be described more, maybe by simulations or by describing its greater applicability. The introduction mentions two broad challenges in the analysis of TS strategy but does not describe at a high level how these challenges are overcome. Due to these shortcomings, I put it below the acceptance threshold.

====== After reading author response, I've increased the score.

---

> ### Comment · Reviewer_TkoG · 2021-11-29
> **After reading author response**
>
> Thank you, the response was clarifying and I increased my score. It would be ideal if the statement "The high level motivation is that the TS-approach has better performance than existing solutions in the most general combinatorial pure exploration setting" was included somewhere in the manuscript.
>
> In Appendix D it is mentioned "In fact, existing optimal learning policies need to use the more powerful offline oracles to efficiently find the base arm that needs exploration the most." But that is only due to computational complexity and not sample complexity, eg, NaiveGapElim of [Chen et al 2017] comes within log factors of the H_{0,c} lower bound. Hence, unless I'm mistaken, the last line in section 2 "However, since we do not assume the existence of a more powerful offline oracle, we cannot achieve the optimal complexity upper bound." still seems inaccurate. It would be better to indicate which type of complexity is being referred to in section 2.

---

> > ### Author Response · Authors · 2021-11-29
> > **Thanks for the update!**
> >
> > Thank you very much for the update on the rating! We're happy to know that you're satisfied with the response. We highly appreciate that.
> >
> > We will include more explanations about **the high level motivation of applying TS-approach in pure exploration problems** in our final version.
> >
> > As for the statement "however, since we do not assume the existence of a more powerful offline oracle, we cannot achieve the optimal complexity upper bound", it is indeed not accurate. Thanks for pointing out it. We will revise it to "however, since we do not assume the existence of a more powerful offline oracle, we cannot achieve the optimal complexity upper bound **efficiently (i.e., with time cost polynomial with $m$)**".

---

### Official Review · Reviewer_MaqF · 2021-11-08

**Correctness:** 3
**Technical Novelty And Significance:** 2
**Empirical Novelty And Significance:** Not applicable
**Recommendation:** 6
**Confidence:** 3

**Main Review:**

Overall I found the problem and proposed method interesting.  Carefully exploiting combinatorial action sets is non-trivial extension of BAI and top-k problems, even for linear joint rewards.  And the proposed verification procedure has some novelty in employing Thompson sampling; Thompson sampling methods have been used for other multi-armed bandit problems, but I am not aware of one in an explore-then-verify procedure.

Overall I think the work is good.  Most of my questions/suggestions I believe can be addressed through writing (eg more verbal justification or clarification)

Motivation for not using UCB-style verification method: I think the explanation in the first two paragraphs on page 2 need to be tightened up -- from the description, it sounds like you can simply rescale the confidence bounds used (since the user knows how many times each base arm is sampled and can construct the UCB for each set of arms to be the specified value).  Also, and not clear what it means for the random sample to be independent --independent with respect to what?  The base arms are from independent samples; do the authors mean that when you compute joint confidence regions for subsets of arms, and the subsets overlap (common base arms), that those mean values (for subsets of arms) will be correlated?  That would make sense to me.

Base arm independence – Are the base arms assumed to be independent? (eg the base arms have diagonal covariance matrix)
Actions – if I understood correctly, only base arms can be pulled.  Is there a reason why the agent cannot pull combinations of arms?  Is that motivated from applications?  Also, would the lower/upper bounds depend on that (e.g. could you achieve better sample complexity by using subsets of arms in the verification step?)

Oracle models – The authors say multiple times that prior works assume powerful oracles, while the authors consider a weaker oracle, emphasizing that point for why the proposed method is only asymptotically optimal.  I don’t think there was any formal comparison about assumptions of the oracle that makes the differences clear, just statements that other papers considered more ‘powerful’ oracles.

Experiments – The work is theoretical in nature, and the authors proved sample complexity bounds.  I think it nonetheless is important to have some empirical results at least for some simple environments; for instance it can confirm that there are not large hidden constants in simple cases.

Minor:
•	Organization suggestion: for readers not familiar with ‘explore-then-verify’ approaches, the word ‘verify’ or ‘verification’ does not appear until the bottom of page two --- it would be helpful to early on explain about why breaking up the pure exploration problem into two parts is helpful.    The authors do describe the basic explore-then-verify algorithm in the appendix; my suggestion here is an earlier, simple verbal explanation in the introduction for why it is a good approach
•	Fact 1 – wouldn’t the lack of termination cause a problem?
•	H_{i,c} notation is properly defined, though a verbal explanation for what they are measuring would be helpful


**Summary Of The Paper:**

This work studies pure exploration for stochastic combinatorial multi-armed bandits, for the fixed confidence setting.  Thus, the goal is to take actions so that the optimal action is identified with high probability (achieving a pre-specified fixed confidence) with as few samples as possible.  While similar problems have been well-studied when action space has (best arm identification, top-k identification), the more general combinatorial setting has not been.  A (common) simplifying assumption of linear reward is used.

This work proposes a explore-then-verify type method.  For some action sets and confidence levels, the explore stage can be done quickly while verification can be challenging.  In the proposed method, base arms are first explored (this work suggests low complexity off-the-shelf method), then a verification stage, for which the method proposes and analyzes a novel verification algorithm using Thompson-sampling.

The authors prove sample complexity bounds for their method, showing it can be competitive against state of the art methods, and is asymptotically optimal (as the confidence level delta goes to 0).


**Summary Of The Review:**

Overall I think the work is good.  I think the contribution is significant enough to merit publication.  Most of my questions/suggestions I believe can be addressed through writing (eg more verbal justification or clarification)  I did not carefully check the proofs, but nothing stood out as suspicious.

---

### Decision · Program_Chairs · 2022-01-20

**Decision:**

Reject

**Comment:**

The reviewers overall thought the problem was worth studying.  However, no reviewer was particularly excited about this work.  The main concern was that the new problem formulation is difficult to compare to prior work. Reviewers felt both more explanation and a deeper detailed comparison would make this a stronger paper.